# "Hurts less, lasts longer"; a qualitative study on experiences of young people receiving high-dose subcutaneous injections of benzathine penicillin G to prevent rheumatic heart disease in New Zealand

Julie Cooper[1], Stephanie L. Enkel[2], Dhevindri Moodley[1], Hazel Dobinson[3], Erik Andersen[3], Joseph H. Kado[2,7], Renae K. Barr[2], Sam Salman [2,4,5], Michael G. Baker[1], Jonathan R. Carapetis[2,6,7], Laurens Manning[2,5,7,8], Anneka Anderson[9], Julie Bennett [1]*

1 Department of Public Health, University of Otago, Wellington, New Zealand, 2 Telethon Kids Institute, Nedlands, Western Australia, Australia, 3 Te Whatu Ora, Capital, Coast and Hutt Valley, Newtown, Wellington, New Zealand, 4 Clinical Pharmacology and Toxicology Unit, PathWest, Perth, Western Australia, Australia, 5 Internal Medicine, The University of Western Australia, Crawley, WA, Australia, 6 Perth Children's Hospital, Nedlands, Western Australia, Australia, 7 Centre for Child Health Research, The University of Western Australia, Crawley, WA, Australia, 8 Fiona Stanley Hospital, Nedlands, Western Australia, Australia, 9 Faculty of Medical and Health Sciences, Te Kupenga Hauora Māori, University of Auckland, Auckland, New Zealand

* julie.bennett@otago.ac.nz

**Data Availability Statement:** Data cannot be shared publicly to align with Kaupapa Māori

## Abstract

### Background

Four-weekly intramuscular (IM) benzathine penicillin G (BPG) injections to prevent acute rheumatic fever (ARF) progression have remained unchanged since 1955. A Phase-I trial in healthy volunteers demonstrated the safety and tolerability of high-dose subcutaneous infusions of BPG which resulted in a much longer effective penicillin exposure, and fewer injections. Here we describe the experiences of young people living with ARF participating in a Phase-II trial of SubCutaneous Injections of BPG (SCIP).

### Methodology

Participants (n = 20) attended a clinic in Wellington, New Zealand (NZ). After a physical examination, participants received 2% lignocaine followed by 13.8mL to 20.7mL of BPG (Bicillin-LA®; determined by weight), into the abdominal subcutaneous tissue. A Kaupapa Māori consistent methodology was used to explore experiences of SCIP, through semi-structured interviews and observations taken during/after the injection, and on days 28 and 70. All interviews were recorded, transcribed verbatim, and thematically analysed.

### Principal findings

Low levels of pain were reported on needle insertion, during and following the injection. Some participants experienced discomfort and bruising on days one and two post dose;

Research and data sovereignty principals and conditions of the study ethical approval. The data presented in this paper is qualitative (interviews with participants) and ethics agreements have said that it must remain confidential. The name of a non-author point of contact regarding data is "June Atkinson" Data Manager, University of Otago, Wellington, June.atkinson@otago.ac.nz. The data behind the pain scores are available as SupportingInformation.

**Funding:** Cure Kids grant 7012. The funders of this study had no role in the study design, data collection and analysis, decision to publish or preparation of the manuscript.

**Competing interests:** The authors have declared that no competing interests exist.

however, the pain was reported to be less severe than their usual IM BPG. Participants were 'relieved' to only need injections quarterly and the majority (95%) reported a preference for SCIP over IM BPG.

## Conclusions

Participants preferred SCIP over their usual regimen, reporting less pain and a preference for the longer time gap between treatments. Recommending SCIP as standard of care for most patients needing long-term prophylaxis has the potential to transform secondary prophylaxis of ARF/RHD in NZ and globally.

## Introduction

Acute rheumatic fever (ARF) is an immune-mediated inflammatory disease that occurs as a delayed sequela to group A streptococcus (GAS) infection [1, 2]. A single acute or several episodes of ARF can progress to rheumatic heart disease (RHD); a serious condition characterised by permanent heart valve damage that may result in early death. It is estimated that 40.5 million people worldwide live with RHD, resulting in over 398,000 deaths each year [3, 4]. ARF and RHD have all but disappeared from high-income countries, yet in Aotearoa New Zealand (NZ), they remain an alarming and inequitable cause of preventable suffering and death for Māori and Pacific Peoples [5].

Since the 1950s, benzathine penicillin G (BPG) has been used extensively for the treatment of several infectious diseases—including ARF and RHD–with a unique and useful characteristic being its prolonged serum concentration. Referred to as secondary prophylaxis [6, 7], four-weekly intramuscular (IM) injections of BPG [8] are needed to ensure plasma penicillin concentrations remain above 0.02mg/L (20ng/mL), a pharmacological surrogate of protection against repeated GAS infections that may worsen disease [9]. Current NZ guidelines recommend patients with ARF have a minimum 10 years of secondary prophylaxis or until the patient is aged 21 years for mild disease, 30 years for moderate or 40 years for severe cases [10]. It is also recommended that patients receive at least 80% of secondary prophylaxis injections (11 of 13 each year); however, reported adherence is much lower than this [11].

The pain and fear associated with IM BPG injection is frequently cited as a reason for lack of adherence to secondary prophylaxis [12, 13]. Therefore, the addition of lignocaine to reduce injection pain has been recommended by the NZ Ministry of Health [14]. Additionally, application of pressure and temperature packs are two methods that have shown significant reduction in pain scores with IM BPG injections [12, 15, 16]. However, ultimately to improve adherence and prevent disease progression there is an urgent need to improve the delivery and formulation of long acting penicillin [17].

A consultation with global experts in RHD suggested changing formulations of BPG to produce a product that is longer acting, less painful, and/or more reliable in its pharmacokinetics [13, 18]. They concluded that an acceptable reformulation would need to be administered subcutaneously; have a dosing schedule greater than six weeks; be less or no more painful than existing BPG; be cold-chain independent; and of comparable cost to IM BPG [13].

Recent work has found subcutaneous (SC) delivery of BPG to be safe and potentially advantageous. In a randomised cross-over trial, Kado et al., [19] compared the pharmacokinetic profile and tolerability of BPG delivered by IM and SC routes of administration. Subcutaneous delivery was superior, having a more prolonged duration of effect, comparable pain scores and equivalent safety [19].

Building on this trial it was predicted that SC infusion of high-dose BPG could provide adequate penicillin concentrations for up to three months. To test the safety and tolerability of high-dose SC infusion of BPG, a dose escalation Phase-I trial was conducted in 24 healthy adult volunteers [20]. The study concluded that delivering high-dose BPG by SC infusion was safe, had acceptable tolerability and could be suitable for up to three-monthly dosing intervals for secondary prophylaxis of ARF/RHD [20]. In addition to the Phase-I trial, a qualitative sub-study was undertaken to provide in-depth information about the tolerability and acceptability of the delivery method [21]. The sub-study demonstrated that the procedure was acceptable to participants, and while some experienced high pain levels, most had tolerable mild discomfort. Potential approaches to alleviate pain and discomfort were explored and suggestions included distractions, a slower infusion time and larger doses of lignocaine anaesthesia.

Incorporating learnings from the Phase-I trial, a Phase-II trial was undertaken to investigate the delivery of high-dose SC Injections of BPG (SCIP) in NZ children and young adults with ARF on secondary prophylaxis. This paper reports the qualitative component of the Phase-II trial, which aimed to explore participants' experiences of SCIP in comparison to their usual intramuscular regimen.

## Methodology

This qualitative study used a Kaupapa Māori consistent methodology [22]. Unlike Kaupapa Māori research that is led by Māori to improve outcomes for Māori through application of Māori autonomy and paradigms [22, 23], Kaupapa Māori consistent approaches are led by non-Māori but align to key Kaupapa principles. Kaupapa Māori research is a critical framework that gives meaning to the life of Māori and analyses unequal relations of power that influence Māori wellbeing. Such a framework allows for an empowering lens that places Māori at the centre of the study and rejects cultural deficit explanations [23, 24]. This study drew on several key principles of Kaupapa Māori methodology including whanaungatanga (family centred methods), te reo (promoting and normalising Māori language), āta (establishing respectful relationships) and taonga tuku iho (incorporating cultural aspirations through prioritising Māori perspectives). Researchers strived to address unequal power relations with participants and maintain a strengths-based approach through cultural safety training and practising self-reflection.

The methods of the Phase-I trial have been described elsewhere [20] with the Phase-II trial following a similar approach. In brief, 37 participants with ARF, identified through a regional register of secondary prophylaxis, were invited to be part of the study, between 21[st] November 2022 and 20[th] March 2023. The study was initially mentioned to potential participants by Māori and Pacific community nurses who administer secondary prophylaxis. For those who said they were interested, the study nurse then phoned to explain the study and sent information packages via email to those who wanted one. The study nurse contacted these potential participants two-weeks after the initial contact and asked if they had any questions. Once all questions were answered, participants were invited to take part in the study, of whom 20 agreed. Participants and any support people were then invited to attend an outpatient clinic in Wellington, NZ. Here the study nurse was able to answer any additional questions and provided hard copies of the participant information pack. Signed consent to participate was given by all participants or Guardians for those under 16 years of age.

Following a physical examination, participants received 2% lignocaine (up to 5mLs) to the subcutaneous abdominal space through a 22G Saf-T-Intima Cannula, inserted by a study nurse specifically trained in this procedure. This was followed by 13.8mL (7.2 million units [MU]) to 20.7mL (10.8 MU) of BPG delivered via a series of slow manual pushes, from

**Table 1. Bicillin-LA dosing for subcutaneous injection.**

| Weight (kg) | Number of pre-filled syringes | Volume (mL) |
|---|---|---|
| 30–50 | 6 | 13.8 |
| 50–70 | 7 | 16.1 |
| 70–90 | 8 | 18.4 |
| >90 | 9 | 20.7 |

manufacturer's prefilled 2.3mL glass syringes (Bicillin-LA®, Pfizer), [25] dosed according to participant's weight (Table 1). One difference in the BPG administration between the Phase-I trial and the Phase-II trial was the use of a spring-driven syringe infusion pump (Springfusor®30, Go Medical Industries Pty Ltd., Subiaco, Australia) with a variable flow control device (VersaRate® Plus, EMED Technologies, El Dorado Hills, California, USA) to deliver Bicillin LA® in the Phase-I trial. The Phase-II trial opted instead to use a series of slow, steady pushes of the Bicillin-LA® pre-filled syringe, as this enabled better control over the injection speed and mitigated the need to transfer the contents into a larger syringe. Participants were followed for 70 days after their initial clinic visit.

## Data collection

Participant observations were undertaken alongside audio-recorded participants interviews, which were conducted at three-time points; day 0 (during and after the injection), day 28, and day 70 following dosing. In addition, participant observations were recorded on days one and two and demographic data were collected via case report forms. All interviews occurred in English face-to-face, with the first taking place at the bedside (JC, JB) in the outpatient department, and the last two in the community were undertaken by the study nurse (generally at the participant's home). Observational data were collected in a field journal by the study nurse (DM) or researcher (JC), transcribed and analysed as described below. The semi-structured interviews used a standardised interview guide consisting of a series of open-ended questions regarding experience of the injection, tolerability of the procedure, pain during and following the injection, and comparisons to their usual IM BPG injections. To supplement the semi-structured interviews, individual audio-recorded informal interviews were had with the study nurse and two researchers who undertook the interviews and were present at the injections. As well as with the community nurse who was responsible for giving monthly IM BPG to participants and who attended several SCIP as a support person.

Pain was measured using a numerical rating score (NRS) on a 0–10 scale during participant interviews: 0 and 10 represented 'no pain' and 'worst pain imaginable', respectively. We considered minimal clinically-important differences (MCID) for moderate pain (NRS 4–7) and severe pain (NRS 8–10) to be 1.3 and 1.8, respectively as reported, although there is no accepted MCID for mild pain (NRS 1–3) [26].

## Data analysis

Interviews and field observations were transcribed verbatim. Pseudonyms are used in place of names. Transcripts and observations were read repeatedly by JC who conducted the majority of the interviews and highlighted initial codes using NVIVO 12 [27]. There was early congruence in themes arising from the transcripts and these were confirmed in discussions between SE, JC and JB. Included in the general inductive data analysis, were observations and experiences captured from those present during the injections. Raw qualitative data codes and themes were presented to a senior Māori researcher (AA) ensuring cultural appropriateness.

### Ethical approval

The ethical considerations of the study were approved by the NZ Health and Disability Ethics Committee (11094) and endorsed by Te Whatu Ora, Capital Coast and Hutt Valley Health, which included review by their Māori Research Board (RAG-M #916). In addition, the trial is registered with the Australian New Zealand Clinical Trials Registry (ACTRN12622000916741). All participants (or their parent / legal guardian) provided written, informed consent, witnessed by a study nurse/researcher, prior to participating. All information and consent forms were available in English, Te Reo Māori and Samoan.

## Results

### Characteristics of participants

Of the 20 participants, the mean age was 15 years (range 7–32). Most participants were female (14, 70%). The majority (12, 60%) identified as Pacific (9 Samoan, 3 Tongan), with seven Māori (35%), and one NZ European (Table 2). On average the injection took 15 minutes (range 9–25 minutes).

### Thematic analysis

The thematic analysis revealed that although there was some initial anxiety arising from fear of the unknown, in general, participants experienced less pain with SCIP delivery when compared to IM. There was a strong preference to remain on SCIP. These findings are further described across six themes below.

**Anxiety in anticipation of SCIP.** Anxiety was either stated or observed in half of the participants. As most of the participants had become used to having their IM BPG injections,

**Table 2. Characteristics of research participants.**

| Pseudonym | Sex | Age (years) | Ethnicity |
|---|---|---|---|
| Leilani | Female | 21 | Samoan |
| Anaru | Male | 15 | Māori/NZ European |
| Matisse | Female | 16 | Samoan |
| Sana | Male | 7 | Samoan |
| Mika | Male | 17 | Samoan |
| Mia | Female | 12 | Samoan |
| Hana | Female | 22 | Māori |
| Ani | Female | 16 | Māori |
| Talia | Female | 21 | Samoan |
| Lucy | Female | 17 | NZ European |
| Mele | Male | 14 | Tongan |
| Aroha | Female | 15 | Māori |
| Sione | Male | 13 | Tongan |
| Lulu | Female | 13 | Samoan |
| Marino | Male | 12 | Māori/NZ European |
| Nina | Female | 32 | Māori |
| Sefina | Female | 13 | Samoan |
| Natia | Female | 12 | Samoan |
| Kiri | Female | 10 | Māori |
| Langi | Female | 10 | Tongan |

coming in for an unknown procedure was an uncomfortable experience, with many participants arriving anxious, as described by Talia and Leilani:

*"A bit nerve wracking. I was a bit anxious. But it was okay, there was no pain."*

*"I thought it would hurt more; I was overthinking it."*

Developing a rapport with SCIP providers early on was key to reducing participant's anxiety. Likewise, offering a *koha* (acknowledgement) in recognition of participant contribution and expenses (travel, parking), plenty of *kai* (food), iPads and Wi-Fi as distractions, and referral of *whānau* (families) to support services when needed, were all reported by participants as key facilitators of rapport and anxiety reduction. As the study progressed the study nurse was also able to reassure participants about some of the positive experiences from the early participants.

**Little pain during the needle insertion.** Over half of participants reported that inserting the needle containing the analgesia was not painful, it was often noted that '*I barely felt it*' or described it as a tickle/tingle. For participants who did feel pain, it was described as a short, sharp, or caused stingy sensation that quickly subsided. Sana and Leilani describe their experiences:

*"I barely felt it, didn't hurt, bit of a pinch"* [from nurse holding skin].

*"Quite sharp pain didn't last very long, just when it goes into the skin then can't feel it after that."*

**Little or no pain during the subcutaneous injection.** Three-quarters of participants described having no pain or minimal pain during the injection. This was validated through the NRS where half of participants reported no pain at all during or after the injection (NRS of 0). It was noted by the nurse and the researchers that many of the participants relaxed once the injection started and many participants became sleepy during the procedure. Hana, Sana and Matisse describe that the injection was not painful:

*"No pain at all, just felt nurses' hand on my tummy, [when needle went in]. During the process I didn't notice it at all to be honest, I could see it but couldn't feel."*

*"No pain"* (smiling)

*"Don't feel anything, no pain"*

However, for some participants the injection was associated with pain, which was often described as stinging, hard or numb. The pain was validated through pain scores which ranged from NRS 1–5. Sione described the injection as:

*"Weird, hot as it was going in. Burning, stinging. It stops and goes. Halfway through pain went up then when finished it went down."*

For a minority the pain increased during the injection from low levels to sometimes quite painful nearing the end. Near the end of the injection Nina described feeling:

*"A little bit uncomfortable now, it is like a pull pain, like the volume expanding."*

Two participants experienced some distress and discomfort, with corresponding pain scores of seven. Without an understanding of participants' pain tolerance, it was hard to know how much of the pain was a result of anxiety to the procedure. By the time they went home they were all feeling more comfortable (pain scores all below three). Aroha explains how she was feeling.

*"It is so sore going in, stingy sore like a bee sting. Oh my god I was crying because I was anxious. I was scared for my life."*

**Experiences and wellbeing in the days and weeks following SCIP.** The Research Nurse observed that over half of participants had some discomfort one day following dosing and some participants had redness or bruising. However, by day two participants reported less pain, mean NRS 1.7 (range 0–5), and by day three all participants were free of discomfort or pain and had resumed usual activities. On day 28 post-dosing when participants reflected on their experience almost all reported that the pain on day one and two had not stopped them doing their usual activities. Matisse found that the injection didn't stop her doing anything:

*"Only a little pain and bruising did not stop me from doing anything."*

**Intramuscular BPG is more painful than SCIP.** When compared to their normal IM injection three-quarters of participants found SCIP significantly less painful, with provided pain scores for their last IM averaging NRS 5 (range 0–9). Following IM injections, several participants noted that the pain post-IM lasted several days and made it uncomfortable to sit down, sleep or participate in normal activities. Talia and Hana describe how they felt after their usual IM injection:

*"With my injection I feel pain which lasts 3–4 days sitting down and sleeping. With this injection one I didn't have that pain"*

*"Pain [from monthly injection] is like a sore muscle like I have been hit with a tennis ball. It is every time I put pressure on it like sitting down and this lasts for two to three days."*

Mele and Matisse also found SCIP less painful than their normal IM injection:

*"I prefer it on my tummy, as it took just takes two days for pain to go away but with the injection in my butt it takes a week as it gets sore every time I sit down"*

*"This one is better [than normal] it does not hurt. My normal injection would be a 5" [Inserting the SCIP needle was rated 1]*

**Strong preference to remain on three-monthly SCIP rather than IM BPG.** Participants were asked on day 28 (day of regular IM BPG) and 70 (day returning to IM BPG) whether they would prefer SCIP or IM BPG. By day 70 all participants (excluding one) said they would prefer to have SCIP. The study nurse observed participants' sense of relief at the prospect of being able to change from IM BPG to SCIP. The key reasons were convenience of having SCIP every three months and reduced pain. Nina and Mia were happy to remain on SCIP:

*"I am super relieved, less admin, one less thing for me to think about."*

*"I'd prefer this injection because I don't have to get injections every month."*

The one participant who preferred not to continue SCIP was a seven-year-old child who tolerated SCIP well but had such a good relationship with the nurse who delivered their regular IM BPG injection that they preferred to see them regularly.

## Discussion

This study described patient perspectives alongside nurse and researcher observations of a Phase-II trial investigating the delivery of SCIP for those currently receiving standard IM BPG treatment. Our results demonstrate a strong preference by participants with ARF presently prescribed four-weekly secondary prophylaxis to receive less frequent delivery of penicillin via a SC injection. Of 20 participants, 19 wished to remain on SCIP rather than return to their regular four-weekly injection. The young participant who preferred to remain on IM BPG tolerated SCIP very well but having formed a good connection with their usual nurse, preferred to continue seeing them regularly. This highlights the importance of whanaungatanga (provider relationships) and child-focussed care for children and young people undergoing repeated painful procedures.

The high level of support for SCIP was due to reported lower pain levels experienced during the injection and while some participants experienced discomfort on days one and two following dosing, the pain reported was less severe and of shorter duration than that of their usual injection. These findings support those from a randomised cross-over trial undertaken by Kado et al., [19] which reported a significantly higher median pain score for those receiving IM BPG (1 [0.25–2]) as opposed to subcutaneous delivery of BPG (0.5 [0–1]), p = 0.03) 48 hours following dosing [19]. The findings are also supported by a qualitative study undertaken in a Phase-I study, which demonstrated that SC infusion of BPG was acceptable to healthy volunteers with most reporting tolerable mild discomfort [21]. In addition to preferring SCIP because of the reduction in pain, participants responded favourably to the longer duration (70–91 days versus 28 days) between dosing. This meant participants has less interruption to their usual work or school schedules and increased freedom to travel.

Administering BPG through a series of slow pushes enabled greater control in the rate of delivery in comparison to the spring infuser used in the Phase-I trial. This potentially led to slightly shorter injection times; Phase-I mean time of 22 minutes, range 16–29 minutes [21], compared to a mean of 15 minutes, range of 9–25 minutes, in Phase-II. In addition, participants in the Phase-II trial reported slightly lower pain scores during the injection in comparison to the Phase-I trial (median NRS 1.0, range 0–7 median compared to median NRS 2.5, range 0–8) [20]. These lower pain scores may reflect that Phase-II participants had a reference of pain associated with IM to compare to. Further improvements to the delivery of SCIP may be achieved by ensuring patients develop a rapport with SCIP providers to help reduce participant anxiety and improve patient experience. Facilitating a shorter injection time, which was typically associated with lower pain and would offer greater convenience for patients and healthcare providers. This could be further supported by penicillin reformulations with a reduced dose volume and/or lower force of administration.

One limitation of the study is that we did not ask about participants usual or general pain tolerance prior to the study. However, we did compare pain experienced during and following SCIP to participants' usual IM BPG (although the latter was reported in retrospect, as a recollection of the pain following their most recent IM BPG injection, which in most cases was four-weeks previously). Another limitation was that as the interviewers were not well known to the young participants, at times it was difficult for the interviewer to obtain in depth feedback on participants experiences and feelings. This may have limited the interviewers understanding of whānau (family) dynamics, which potentially altered the content of feedback.

However, we experienced that whānau were very supportive in terms of helping younger participants provide examples of experiences particularly for those participants who found it hard to share their experiences.

A strength of this study was the high level of engagement from participants, with 100% adherence. This was reflective of trust, communication whanaungatanga, and rapport developed between the study team, health practitioners, and participants. However, this relationship building may have also led to a response bias, with participants wanting to provide favourable responses, although the study used open-ended nature of the interview questions meant participants were able to discuss challenges with regular IM BPG and day-to-day life. In addition, the use of the a Kaupapa Māori consistent approach as a methodology helped participants to feel empowered and treated as partners rather than as subjects of research.

SCIP utilising pre-filled syringes is preferred over IM BPG as a tolerable and acceptable route of administration for children and young adults with ARF in Wellington, NZ. Extending this study to support widespread use of SCIP throughout NZ in patients requiring secondary prophylaxis over a prolonged period will enable better assessment of key measures over time. These measures should investigate preference for SCIP outside of the study region and confirm that SCIP will facilitate improved adherence to secondary prophylaxis. We note that SCIP does not align with all recommendations of global experts in that it should be cold chain independent and of comparable cost for greatest global benefit. It would be useful to undertake an economic evaluation of SCIP in comparison to IM BPG. Further work on cold chain independence is warranted, potentially exploring whether reconstituted, lyophilised formations, as are used in low- and middle- income countries, are able to be delivered through a SCIP approach. On the assumption that SCIP continues to demonstrate favourable pharmacokinetics, the evidence from this and other studies looking at SC delivery of BPG suggest that SCIP shows enormous promise as an alternative mode of delivery for patients needing long-term prophylaxis. SCIP has the potential to improve adherence through improved patient experiences, which may prevent disease progression and death, transforming prophylaxis of ARF/RHD both in NZ and globally.

## Supporting information

**S1 File. Pain scores raw data.**
(XLSX)

## Acknowledgments

We would like to thank all the participants and their *whānau* (families) for sharing their time and experiences to make this study possible. We would also like to thank the outpatient department at Kenepuru Hospital for sharing their space with us and the wonderful community nurses for supporting the study.

## Author Contributions

**Conceptualization:** Joseph H. Kado, Renae K. Barr, Sam Salman, Laurens Manning, Julie Bennett.

**Data curation:** Julie Cooper, Julie Bennett.

**Formal analysis:** Stephanie L. Enkel, Julie Bennett.

**Funding acquisition:** Hazel Dobinson, Joseph H. Kado, Renae K. Barr, Sam Salman, Michael G. Baker, Jonathan R. Carapetis, Laurens Manning, Anneka Anderson, Julie Bennett.

**Methodology:** Stephanie L. Enkel, Joseph H. Kado, Sam Salman, Laurens Manning, Anneka Anderson, Julie Bennett.

**Project administration:** Dhevindri Moodley, Julie Bennett.

**Resources:** Hazel Dobinson, Erik Andersen, Julie Bennett.

**Supervision:** Anneka Anderson, Julie Bennett.

**Visualization:** Dhevindri Moodley, Laurens Manning, Julie Bennett.

**Writing – original draft:** Julie Cooper, Julie Bennett.

**Writing – review & editing:** Stephanie L. Enkel, Dhevindri Moodley, Hazel Dobinson, Erik Andersen, Joseph H. Kado, Renae K. Barr, Sam Salman, Michael G. Baker, Jonathan R. Carapetis, Laurens Manning, Anneka Anderson, Julie Bennett.

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
