## [Decision Letter · Decision Letter 0]

26 Dec 2023

PONE-D-23-38244“Hurts less, lasts longer”  experiences of young people receiving high-dose subcutaneous infusions of benzathine penicillin G to prevent rheumatic heart diseasePLOS ONE

Dear Dr. Bennett,

Thank you for submitting your manuscript to PLOS ONE. After careful consideration, we feel that it has merit but does not fully meet PLOS ONE’s publication criteria as it currently stands. Therefore, we invite you to submit a revised version of the manuscript that addresses the points raised during the review process. Please submit your revised manuscript by Feb 09 2024 11:59PM. If you will need more time than this to complete your revisions, please reply to this message or contact the journal office at plosone@plos.org. Please include the following items when submitting your revised manuscript:A rebuttal letter that responds to each point raised by the academic editor and reviewer(s). You should upload this letter as a separate file labeled 'Response to Reviewers'.A marked-up copy of your manuscript that highlights changes made to the original version. You should upload this as a separate file labeled 'Revised Manuscript with Track Changes'.An unmarked version of your revised paper without tracked changes. You should upload this as a separate file labeled 'Manuscript'.If applicable, we recommend that you deposit your laboratory protocols in protocols.io to enhance the reproducibility of your results. Protocols.io assigns your protocol its own identifier (DOI) so that it can be cited independently in the future. For instructions see: https://journals.plos.org/plosone/s/submission-guidelines#loc-laboratory-protocols. Additionally, PLOS ONE offers an option for publishing peer-reviewed Lab Protocol articles, which describe protocols hosted on protocols.io. Read more information on sharing protocols at https://plos.org/protocols?utm_medium=editorial-email&utm_source=authorletters&utm_campaign=protocols.

We look forward to receiving your revised manuscript.

Kind regards,

Stephen Esaku

Academic Editor

PLOS ONE

Journal Requirements:

"Cure Kids grant 7012"

Reviewers' comments:

Reviewer's Responses to Questions

**Comments to the Author**

1. Is the manuscript technically sound, and do the data support the conclusions?

Reviewer #1: Partly

Reviewer #2: Yes

Reviewer #3: Yes

2. Has the statistical analysis been performed appropriately and rigorously? 

Reviewer #1: N/A

Reviewer #2: N/A

Reviewer #3: N/A

3. Have the authors made all data underlying the findings in their manuscript fully available?

Reviewer #1: Yes

Reviewer #2: No

Reviewer #3: Yes

4. Is the manuscript presented in an intelligible fashion and written in standard English?

Reviewer #1: Yes

Reviewer #2: Yes

Reviewer #3: Yes

5. Review Comments to the Author

Reviewer #1: “Hurts less, lasts longer”: experiences of young people receiving high-dose subcutaneous infusions of benzathine penicillin G to prevent rheumatic heart disease.

Overall Summation:

This study describes a welcome and important innovation considering the length of time that the only option for secondary prophylaxis has been via deep intramuscular injection, with its associated problem of pain. The paper reports on the qualitative component of a Phase-II medication trial. It explores experiences of the new medication formulation among the recruited participants.

The importance and innovative nature of this study is currently let down by its non-alignment with standards in the reporting of qualitative research, such as SRQR. A review to align the paper with guidelines will greatly improve it and ensure that readers perceive that the study is rigorous, believable, and generalisable.

The study was conducted with participants who are Indigenous and, in this case, are impacted by a condition known to arise in settings of socioeconomic disadvantage, power and opportunity within a nation boasting a high standard of living, Aotearoa. The need to use Indigenous methodologies in conducting studies in such settings is a given. I am sure that the study was conducted ethically in this regard, but the current writing does not demonstrate this well enough to the reader. The researchers have utilised Kaupapa principles. This has been given a “passing nod” that is not convincing and is one place where the paper should be strengthened. As recommended in SRQR guidelines, the Indigenous status of researchers, including those who conducted interviews and observations needs to be made clear.

Pain and frequency of the usual SP regimen are presented as the only limiting factors impacting adherence to the injection regimen. As I am sure the authors are aware, these are just some of the limiting factors. See synopsis for a more accurate description to do with adherence.

One of the strongest outcomes of this trial is the potential to provide another SP option for patients to choose from. Query as to what happened at the end of the trial to the 19 participants who strongly preferred SCIP needs to be included.

Title:

The title should clearly show the genre of the research and the study site.

Suggestions:

“Hurts less, lasts longer”: a qualitative study on experiences of young people receiving high-dose subcutaneous infusions of benzathine penicillin G to prevent rheumatic heart disease in Aotearoa (or New Zealand).

Or

A qualitative study on experiences of young people receiving high-dose subcutaneous infusions of benzathine penicillin G in a Phase-II trial to prevent rheumatic heart disease in Aotearoa (or New Zealand): “hurts less, lasts longer”.

Abstract:

Line 7 Methodology: Need to include “methodology”. Currently just states methods (interviews and observations- and the trial process).

Suggest adding at the start of this paragraph:

“This study used Kaupapa principles to explore Māori and Pacific young peoples’ experiences.”

Line 17 Remove the emotive word “overwhelmingly” to fit with academic writing style.

Line 19 Suggest replacing “treatments” with injections.

No matter which method patients choose, it is still an injection.

Synopsis:

In general, A solely biomedical description of ARF and RHD in a qualitative paper. This likely reflects the fields of expertise of most of the authors. Suggestions below

Line 25 Suggest inserting the following text at the end of the sentence after the word infection:

“…predominantly among people living in socioeconomic disadvantage.”

Line 29 Needs a more accurate description to do with adherence problems. Change sentence to:

“The effectiveness of this approach is limited by complex issues including the young age of patients and health system frailties (as in your ref 8). However, pain, fear, and the frequency of injection are recognised as common barriers to the injection adherence."

Line 34 Replace “overwhelmingly” with strongly

Line 35-36 Put the number 70 first to match with the preceding text

Introduction:

Line 48 “…for Indigenous Māori”.

Debates exist around use of this term in Aotearoa. Could distract some readers. Note that in ref 4, some of the same authors as in this paper use the phrase “people of Māori and Pacific Islander ethnicity”.

Line 58 Typo- recommend should be recommended

Line 62 Change beginning of sentence to:

“Pain and fear associated with IM…….”

Line 67 New formulations are urgent but adherence to the regimen is more complex than just injection pain and there is an equally urgent need, for instance, to improve housing in NZ for Māori and Pacific people to prevent ARF.

Suggestions for new sentence:

However, to improve adherence and prevent disease progression, new formulations of long-acting penicillins are warranted.

or

Improving the formulation of long-acting penicillin has potential to improve adherence and prevent disease progression.

Line 73 and 74 For later ref in last paragraph of the Discussion, need to mention the global experts’ conclusions about cold chain and cost mentioned here.

Line 79 End of sentence change to “no significant adverse effects”

Line 89 and 90 Note: These lines show the additional measures that should be available to address pain and fear of injection episodes for young people thus providing evidence that the authors do know that the injection experiences are more complex than pain alone.

Line 97 Suggest change to:

This paper reports on the qualitative component of the SCIP-II trial aiming to explore participants’ experiences of SCIP compared with the usual intramuscular regimen.

Methods

Further information that aligns with the Kaupapa principles is needed in describing recruitment of participants.

Line 93 Suggested rewrite:

“In brief, participants with ARF, were identified through a regional register of secondary prophylaxis. X were invited to be part of the study by attending an outpatient clinic (designated for the study?) in Wellington, NZ.”

A brief statement of how the participants were invited is needed.

Were they contacted by phone, email or in person? And by whom? How were explanations of the study provided? This info is needed to demonstrate a Kaupapa approach of respecting participants’ equal power to choose to take part in the study or not.

Finally,

“Twenty participants agreed to participate in the study which ran from 21st November 2022 until 20th March 2023. Participants were followed for 70 days after their initial clinic visit where they received SCIP.”

Table line 106 For clarity, change “Number of vials” to “Number of prefilled syringes”

Line 110 Change data collection here to a new section heading, “Methodology”.

This new Methodology section would assist flow if it were placed before the Methods section because it provides rationale for why certain methods were undertaken.

Stating the methodology strengthens qualitative papers and is one indicator of quality of a study. Not being clear about methodology is a common trap in qualitative components of drug trials.

Suggested text below

Methodology

Line 111 Suggested example to replace existing text on Kaupapa principles in research. Current explanations of Kaupapa in the paper are vague and unconvincing for the reader:

“This qualitative study used Kaupapa Māori research principles as a framework. These principles prioritise Māori perspectives and wellbeing by attending to unequal power relations and rejecting deficit explanations.”

This is where you need to state that the study involved Māori or Pacific researchers, nurses of assistants.

How were participants given feedback on results? Did they know upfront that it was a trial and that SCIP would not be available after the study ended? How did you mitigate a sense of being experimented on?

Line 116 This subheading of “data collection” describes methods. Suggest moving it to Methods section, below the table 1. Commence with Participant observations….

Note, interviews and observations are methods, not methodology.

Text to do with Kaupapa now under Methodology.

Line 120 All interviews occurred face to face by whom? Initials of researcher suffice.

I note in the ethics section that consent forms were available in Te Reo Māori and Samoan. This implies that language was an issue?? What languages were the interviews conducted in? Were interpreters needed or used?

Line 122 Observational data were collected by the study nurse or researcher. Insert their initials. Hopefully we know that they are Māori or Pacifica or not by this point in the paper to reflect Kaupapa principles and transparency.

Line 127 Ambiguous and needs more detail about:

‘a community nurse’ what is her role and involvement with SP or other?

‘two researchers’: which ones? Were they clinicians, qualitative researchers, authors on this paper? Convince the readers that these interviews were relevant and added value.

Line 129 Remove ‘quantitative’, the pain was measured quantitatively.

Quantitative is implied in the scoring system used.

Qualitative studies can also include quantitative methods.

Line 132 Insert “a reduction of” 1.3 and 1.8 respectively….

Data analysis

Line 137 Change to: Interviews and field observations were transcribed verbatim for analysis.

Line 138 Transcripts were read repeatedly by [insert initials] and remove “one investigator”

Line 139 Thematic saturation is a given in qual research. This sentence and the following sentence could be removed. Better to state that there was early congruence in themes arising from transcripts.

Line 142 Here we finally hear that the study nurse, a community nurse, and two researchers were present during infusions. This info needs to come in line 127.

This sentence not needed here as it is previously explained that they were interviewees.

Line 144 and 145 This is an important sentence that needs further information to verify the Kaupapa framework. The codes and themes were presented to Māori and Pacific researchers (the first time we hear such researchers were involved). We need to hear their responses. Did they adjust any codes, make suggestions, provide insight into cultural aspects? Additionally, was any feedback given to participants? Did this lead to any changes in themes?

Ethical approval

Line 153 Were the consent forms also available in English?

Results

Lines 159 and 160 Age range states 6 -32 but table has youngest as 7 years.

Suggest remove “Indigenous”

Thematic analysis

Line 165 Suggest: “Overall, the thematic analysis………

Some initial anxiety from participants to with… what? Fear of unknown? Dealing with unfamiliar practitioners?

Line 171 Change to “Anxiety prior to SCIP” or “Anxiety in anticipation of SCIP”

Line 172 How was anxiety evident? “Anxiety was either stated or observed in many participants.”

Line 189 …short, sharp or caused a stinging sensation that quickly subsided.

Line 196 …participants described having no pain or minimal pain during……

Line 212 Tautology: don’t need small with minority

Line 216 “Two participants experienced some distress and discomfort, with corresponding pain scores of seven.” Without a prior understanding of ……

Line 223 Move this sentence to Discussion.

Line 231 “…had resumed usual activities.”

Line 255 Change heading to: Strong preference to remain on three-monthly SCIP rather than BPG

Line 257

Line 258 No need for brackets

Suggest change to:

“The study nurse observed participants’ sense of relief at the prospect of being able to change from IM BPG to SCIP”.

This is why readers need to know if it was explained to participants at the start of the study if ongoing SCIP would be an option at the end of the study. It was predictable that if a strong preference emerged for SCIP that this question about continuing SCIP would arise among participants.

Line 266 “…who tolerated SCIP well

Discussion

Line 273 Replace overwhelmingly with strong

Line 276 Suggest change to:

“The seven-year-old participant who chose a known provider over SCIP highlights the importance of provider relationships and child-focused care for children undergoing repeated painful procedures such as SP” (ref new Australian RHD guidelines).

Line 279 …due to ‘reported’ lower pain levels…

Pain is still a subjective experience

Line 282 Replace reinforce with support

Line 286 Replace last sentence:

This meant that participants had less interruption to their usual work or school schedules to obtain injections and increased their freedom to travel.

Line 289 Suggest change to:

“Administering SCIP through a series of slow pushes enabled more control in the rate of delivery in comparison to the ……”

Line 301 Change to…we did not ask about “participants usual or general pain tolerance prior to the study”.

Line 302 Remove the words “and contrast”

Line 304 to 309 These sentences are a red flag to qualitative researchers.

As it currently reads, there are negative assumptions that I doubt the authors intend about some participants and which defy Kaupapa principles.

Please review these sentences with Māori researchers for insight and clarity.

For instance, were the difficulties due to language or cultural barriers or unknown interviewer?

Line 313 Suggest addition:

“…. participants were able to discuss life issues as well as challenges with regular IM BPG indicating their preference for holistic approaches in healthcare delivery”

Line 314 to 316 I think you mean that use of the Kaupapa approach as an Indigenous methodology helped participants to feel empowered and treated as partners rather than subjects of research.

This sentence needs changing.

Line 318 to 324 This study reports a great breakthrough in SP. However, we need to be mindful that the current formulation of SCIP is only possible where there is a cold chain and where it can be afforded. This indicates that more work needs to be done for those sites that don’t have these advantages. Suggest a reword:

“SCIP utilising prefilled syringes is preferred over IM BPG as a tolerable and acceptable route of administration for children and young adults with ARF in Wellington, NZ. Extending this study to support widespread choice of SCIP throughout NZ for patients requiring secondary prophylaxis over a prolonged period will enable better assessment of key measures over time. We note that SCIP does not yet align with all recommendations of global experts in that it should be cold chain independent and of comparable cost for greatest global benefit. It would be useful to undertake an economic evaluation of SCIP in comparison to IM BPG. Further work on cold chain independence is warranted. Current evidence suggests that SCIP will likely become the standard of care for most patients needing long-term prophylaxis. SCIP has the potential to improve patient experiences which may assist prevention of disease progression and death, transforming prophylaxis of ARF/RHD both in NZ and globally.

Reviewer #2: This is a well written and easy to read qualitative study about a potentially important new strategy to managing secondary prevention of ARF.

Whilst the authors state all data is available, I am unable to find a link to it.

The study uses a design designed to empower Maori population - and I think this is important to the study. But I wonder how this fits with the other ethnicities within the study - predominantly Pacific. I'm not sure if this is worth commenting upon.

A few questions regarding methodology for clarification:

- The method of administration could be described in more detail - particularly as it differed to previous studies - multiple slow manual pushes - was this through a single needle pre-inserted subcutaneously? Was this following administration of local anaesthetic? Was training to nurses required for this?

Comments regarding results and discussion:

Whilst I understand this is a quantitative study, a few more numbers would help clarify the results in the thematic analysis.

eg better defining - 'anxiety was present for 'many' participants', and "for 'most' participants inserting the needle containing the analgesia did not cause a lot of pain. I think quantifying this could be helpful - eg x/y participants report anxiety....

The results note that techniques were used to allay anxiety - these may have also influenced pain perception. Are these techniques used when receiving IM BPG also? If not and comparing pain perception between the two methods of administration, this needs to be acknowledged.

Patients knowing they are in a study, particularly if they have a good rapport with the questioner and feel buoyed by being included in the study may lead to response bias - ie wanting to provide a favourable. This could perhaps be acknowledge or explored in the discussion.

In the discussion the sentence on line 319 "Extending this study to support widespread use of SCIP, through NZ in patients requiring secondary prophylaxis over a prolonged period will enable better assessment of key measures." I think this sentence is a little too vague - it will be important to confirm ongoing preference outside the study setting and to confirm this method facilates easier/improved adherence to prophylaxis regime and efficacy.

Reviewer #3: Congratulations for a great work—impressive methodology, well-articulated. The study's robust setup and positive engagement with the RHD clinic and communities are commendable. The only drawback, a small sample size, prompts anticipation for future research on a larger scale. I look forward to witnessing your impactful contributions on a broader spectrum.

6. PLOS authors have the option to publish the peer review history of their article (what does this mean?). If published, this will include your full peer review and any attached files.

Reviewer #1: **Yes: **Alice Mitchell

Reviewer #2: No

Reviewer #3: **Yes: **Subhrajit Lahiri MD

---

## [Author Response · Author response to Decision Letter 0]

25 Feb 2024

For the Editor,

We have added the Funding Statement into the Cover Letter. We have hopefully updated the formatting to align with PLOS One (in the clean version). 

We are unable to share data due to Kaupapa Māori principals and data sovereignty. As well as aligning to our ethics approvals imposed by the Health and Disability Ethics Committee (application: 11094). 

We have added reference 22 below in response to Reviewer comments. The below reference by Watkins is now replaced by Roth et al and Vaduganathan M as these are more up-to-date RHD estimates.

 Watkins DA, Johnson CO, Colquhoun SM, Karthikeyan G, Beaton A, Bukhman G, et al. Global, Regional, and National Burden of Rheumatic Heart Disease, 1990-2015. N Engl J Med. 2017;377(8):713-22. Epub 2017/08/24. doi: 10.1056/NEJMoa1603693. PubMed PMID: 28834488.

Is now replaced by 

3. Roth GA, Mensah GA, Johnson CO, Addolorato G, Ammirati E, Baddour LM, et al. Global Burden of Cardiovascular Diseases and Risk Factors, 1990–2019: Update From the GBD 2019 Study. Journal of the American College of Cardiology. 2020;76(25):2982-3021. doi: https://doi.org/10.1016/j.jacc.2020.11.010.

4. Vaduganathan M, Mensah GA, Turco JV, Fuster V, Roth GA. The Global Burden of Cardiovascular Diseases and Risk: A Compass for Future Health. J Am Coll Cardiol. 2022;80(25):2361-71. Epub 20221109. doi: 10.1016/j.jacc.2022.11.005. PubMed PMID: 36368511.

22. Malpas P, Anderson A, Wade J, Wharemate R, Paul D, Jacobs P, et al. A critical exploration of a collaborative Kaupapa Māori consistent research project on physician-assisted dying. The New Zealand medical journal. 2017;130(1454):47-54. Epub 20170428. PubMed PMID: 28449016.

 

Reviewer 1:

This study describes a welcome and important innovation considering the length of time that the only option for secondary prophylaxis has been via deep intramuscular injection, with its associated problem of pain. The paper reports on the qualitative component of a Phase-II medication trial. It explores experiences of the new medication formulation among the recruited participants. 

The importance and innovative nature of this study is currently let down by its non-alignment with standards in the reporting of qualitative research, such as SRQR. A review to align the paper with guidelines will greatly improve it and ensure that readers perceive that the study is rigorous, believable, and generalisable. 

The study was conducted with participants who are Indigenous and, in this case, are impacted by a condition known to arise in settings of socioeconomic disadvantage, power and opportunity within a nation boasting a high standard of living, Aotearoa. The need to use Indigenous methodologies in conducting studies in such settings is a given. I am sure that the study was conducted ethically in this regard, but the current writing does not demonstrate this well enough to the reader. The researchers have utilised Kaupapa principles. This has been given a “passing nod” that is not convincing and is one place where the paper should be strengthened. As recommended in SRQR guidelines, the Indigenous status of researchers, including those who conducted interviews and observations needs to be made clear. 

Pain and frequency of the usual SP regimen are presented as the only limiting factors impacting adherence to the injection regimen. As I am sure the authors are aware, these are just some of the limiting factors. See synopsis for a more accurate description to do with adherence. 

One of the strongest outcomes of this trial is the potential to provide another SP option for patients to choose from. Query as to what happened at the end of the trial to the 19 participants who strongly preferred SCIP needs to be included. 

Thank you for taking the time to review our manuscript. We have addressed your comments as outlined in the responses below and believe the manuscript is much improved thanks to your input.

 

Title: 

The title should clearly show the genre of the research and the study site. 

Suggestions: 

“Hurts less, lasts longer”: a qualitative study on experiences of young people receiving highdose subcutaneous infusions of benzathine penicillin G to prevent rheumatic heart disease in Aotearoa (or New Zealand). 

Or 

A qualitative study on experiences of young people receiving high-dose subcutaneous infusions of benzathine penicillin G in a Phase-II trial to prevent rheumatic heart disease in Aotearoa (or New Zealand): “hurts less, lasts longer”. 

Thank you for this suggestion, we have updated the title.

Abstract: 

Line 7 

Methodology: Need to include “methodology”. Currently just states methods (interviews and observations- and the trial process). Suggest adding at the start of this paragraph: 

“This study used Kaupapa principles to explore Māori and Pacific young peoples’ experiences.” Unfortunately, the word limit made it difficult to include everything we would have liked to. However, we have added in part of your suggested sentence.

Line 17 Remove the emotive word “overwhelmingly” to fit with academic writing style. This has been done.

Line 19 Suggest replacing “treatments” with injections. 

No matter which method patients choose, it is still an injection. SCIP is an infusion rather than an injection, so the use of the word injection isn’t appropriate here.

 

Synopsis: 

In general, 

A solely biomedical description of ARF and RHD in a qualitative paper. This likely reflects the fields of expertise of most of the authors. Suggestions below 

Line 25 Suggest inserting the following text at the end of the sentence after the word infection: 

“…predominantly among people living in socioeconomic disadvantage.” This has been done.

Line 29 Needs a more accurate description to do with adherence problems. Change sentence to: 

“The effectiveness of this approach is limited by complex issues including the young age of patients and health system frailties (as in your ref 8). However, pain, fear, and the frequency of injection are recognised as common barriers to the injection adherence." Thanks. This has been done.

Line 34 Replace “overwhelmingly” with strongly We have removed overwhelmingly.

Line 3536 Put the number 70 first to match with the preceding text This has been done.

Introduction: 

Line 48 “…for Indigenous Māori”. 

Debates exist around use of this term in Aotearoa. Could distract some readers. Note that in ref 4, some of the same authors as in this paper use the phrase “people of Māori and Pacific Islander ethnicity”. Yes, different people have various views on this. We tend to use Indigenous if the audience is international. However, in this case we have removed “Indigenous”. 

Line 58 Typo- recommend should be recommended Thanks for picking up on this. It has been corrected.

Line 62 Change beginning of sentence to: 

“Pain and fear associated with IM…….” “and fear” has been added to the sentence.

Line 67 New formulations are urgent but adherence to the regimen is more complex than just injection pain and there is an equally urgent need, for instance, to improve housing in NZ for Māori and Pacific people to prevent ARF. 

Suggestions for new sentence: 

However, to improve adherence and prevent disease progression, new formulations of long-acting penicillins are warranted. 

or 

Improving the formulation of long-acting penicillin has potential to improve adherence and prevent disease progression. Poor quality housing in Aotearoa is a massive issue and one that should be addressed not just to prevent ARF/RHD but for many other health related issues, i.e. respiratory/circulatory issues.

The development of ARF is extremely complex as is any primordial prevention intervention. This manuscript is about “secondary prevention” preventing those with ARF getting RHD (heart failure). To solve ARF you need to work along the disease pathway and each area is important. Of course, the ideal is for no one to get ARF. Better quality housing alone may not do this. Research shows that household over-crowding is a risk factor for ARF, but there are other risk factors such as access to primary care. 

Line 73 and 74 For later ref in last paragraph of the Discussion, need to mention the global experts’ conclusions about cold chain and cost mentioned here. Cold-chain and cost have now been mentioned in the discussion.

Line 79 End of sentence change to “no significant adverse effects” This has been done.

Line 89 and 90 

Note: These lines show the additional measures that should be available to address pain and fear of injection episodes for young people thus providing evidence that the authors do know that the injection experiences are more complex than pain alone. 

Line 97 Suggest change to: 

This paper reports on the qualitative component of the SCIP-II trial aiming to explore participants’ experiences of SCIP compared with the usual intramuscular regimen. This has been done.

 

Methods 

Further information that aligns with the Kaupapa principles is needed in describing recruitment of participants. 

Line 93 Suggested rewrite: 

“In brief, participants with ARF, were identified through a regional register of secondary prophylaxis. X were invited to be part of the study by attending an 

outpatient clinic (designated for the study?) in Wellington, NZ.” A brief statement of how the participants were invited is needed. 

Were they contacted by phone, email or in person? And by whom? How were explanations of the study provided? This info is needed to demonstrate a Kaupapa approach of respecting participants’ equal power to choose to take part in the study or not. Finally, 

“Twenty participants agreed to participate in the study which ran from 21st November 2022 until 20th March 2023. Participants were followed for 70 days after their initial clinic visit where they received SCIP.” Potential participants were initially contacted by phone. The study nurse explained the study to those who were interested. If they were still interested, she asked if they would like a participant information pack which she emailed (or posted) to them. Two weeks later the study nurse would call again to ask if potential participants/whanau had any questions. Once these were answered, potential participants were invited to attend an outpatient clinic. Those that attended the clinic were given participant information sheets and provided written consent. 

We have added more information about how participants were recruited into the methods. 

Table line 106 For clarity, change “Number of vials” to “Number of prefilled syringes” 

 This has been done. In addition, anything that mentions vials has been removed.

Line 110 Change data collection here to a new section heading, “Methodology”. 

This new Methodology section would assist flow if it were placed before the Methods section because it provides rationale for why certain methods were undertaken. We have renamed “data collection” as “methodology” and rearranged the section, which hopefully addresses your concerns. We note however, that the journal will have the final decision on these titles to fit with their submission guidelines. 

Stating the methodology strengthens qualitative papers and is one indicator of quality of a study. Not being clear about methodology is a common trap in qualitative components of drug trials. Suggested text below 

Methodology 

Line 111 Suggested example to replace existing text on Kaupapa principles in research. Current explanations of Kaupapa in the paper are vague and unconvincing for the reader: 

“This qualitative study used Kaupapa Māori research principles as a framework. These principles prioritise Māori perspectives and wellbeing by attending to unequal power relations and rejecting deficit explanations.” 

This is where you need to state that the study involved Māori or Pacific researchers, nurses of assistants. 

How were participants given feedback on results? Did they know upfront that it was a trial and that SCIP would not be available after the study ended? How did you mitigate a sense of being experimented on? Thank you for your suggested wording. This has been updated in the text. 

As per the Kaupapa Maori framework, participants and their whanau were active participants in the clinical trial. We tried to ensure engagement with the kaupapa/reasoning for the trial, which mitigated any sense of being experimented on. 

The participant were informed that if the data was positive, there would be an opportunity to continue with the SCIP therapy. The outcomes have now allowed us to offer ongoing SCIP to those who would like it. 

Line 116 This subheading of “data collection” describes methods. Suggest moving it to Methods section, below the table 1. Commence with Participant observations…. 

Note, interviews and observations are methods, not methodology. 

Text to do with Kaupapa now under Methodology. This has been done.

Line 120 All interviews occurred face to face by whom? Initials of researcher suffice. I note in the ethics section that consent forms were available in Te Reo Māori and Samoan. This implies that language was an issue?? What languages were the interviews conducted in? Were interpreters needed or used? Initials of the interviewer have been added to the text. We did have information forms available in Te Reo and Samoan, however, only one whanau requested a Samoan version, which was for an elderly family member. Interpreters were also available, but no one needed/or wanted one. We have updated this section to reflect that all interviews were conducted in English. 

Line 122 Observational data were collected by the study nurse or researcher. Insert their initials. Hopefully we know that they are Māori or Pacifica or not by this point in the paper to reflect Kaupapa principles and transparency. Initials have been added.

Line 127 Ambiguous and needs more detail about: 

‘a community nurse’ what is her role and involvement with SP or other? ‘two researchers’: which ones? Were they clinicians, qualitative researchers, authors on this paper? Convince the readers that these interviews were relevant and added value. The community nurse was responsible for giving IM BPG to patients in the region. She attended several SCIP procedures to support her patients who chose to take part in the study. 

The researchers were present at the SCIP infusions and undertook the interviews. 

Further details have been added to the manuscript to clarify the roles and the value that their voices added. 

Line 129 Remove ‘quantitative’, the pain was measured quantitatively. 

Quantitative is implied in the scoring system used. 

Qualitative studies can also include quantitative methods. This has been done.

Line 132 Insert “a reduction of” 1.3 and 1.8 respectively…. The 1.3 refers to moderate pain and the 1.8 refers to severe pain, therefore there is not a reduction in pain.

Data analysis 

Line 137 Change to: Interviews and field observations were transcribed verbatim for analysis. This has been done.

Line 138 Transcripts were read repeatedly by [insert initials] and remove “one investigator” This has been done.

Line 139 Thematic saturation is a given in qual research. This sentence and the following sentence could be removed. Better to state that there was early congruence in themes arising from transcripts. These sentences have been deleted.

Line 142 Here we finally hear that the study nurse, a community nurse, and two researchers were present during infusions. This info needs to come in line 127. This sentence not needed here as it is previously explained that they were interviewees. This has been removed from here and is highlighted earlier on as mentioned above.

Line 144 and 145 This is an important sentence that needs further information to verify the 

Kaupapa framework. The codes and themes were presented to Māori and Pacific researchers (the first time we hear such researchers were involved). We need to hear their responses. Did they adjust any codes, make suggestions, provide insight into cultural aspects? Additionally, was any feedback given to participants? Did this lead to any changes in themes? This sentence has been amended.

Ethical approval 

Line 153 Were the consent f

---

## [Decision Letter · Decision Letter 1]

5 Apr 2024

“Hurts less, lasts longer”;

a qualitative study on experiences of young people receiving high-dose subcutaneous infusions of benzathine penicillin G to prevent rheumatic heart disease in New Zealand

PONE-D-23-38244R1

Dear Dr. BENNETT,

We’re pleased to inform you that your manuscript has been judged scientifically suitable for publication and will be formally accepted for publication once it meets all outstanding technical requirements.

Kind regards,

Stephen Esaku

Academic Editor

PLOS ONE

Additional Editor Comments (optional):

Reviewers' comments:

Reviewer's Responses to Questions

**Comments to the Author**

1. If the authors have adequately addressed your comments raised in a previous round of review and you feel that this manuscript is now acceptable for publication, you may indicate that here to bypass the “Comments to the Author” section, enter your conflict of interest statement in the “Confidential to Editor” section, and submit your "Accept" recommendation.

Reviewer #1: All comments have been addressed

Reviewer #2: All comments have been addressed

2. Is the manuscript technically sound, and do the data support the conclusions?

Reviewer #1: Yes

Reviewer #2: Yes

3. Has the statistical analysis been performed appropriately and rigorously? 

Reviewer #1: N/A

Reviewer #2: N/A

4. Have the authors made all data underlying the findings in their manuscript fully available?

Reviewer #1: Yes

Reviewer #2: No

5. Is the manuscript presented in an intelligible fashion and written in standard English?

Reviewer #1: Yes

Reviewer #2: Yes

6. Review Comments to the Author

Reviewer #1: (No Response)

Reviewer #2: Very minor comments only:

Line 223 and 242 I think should both be written as past tense. eg

Han, Sara and Matisse described that the infusion was not painful", "Aroha explained how she was feeling"

The names quoted through the paper - have they been changed to protect anonymity? (I apologise if this is stated and I missed it.)

7. PLOS authors have the option to publish the peer review history of their article (what does this mean?). If published, this will include your full peer review and any attached files.

Reviewer #1: **Yes: **Dr Alice G Mitchell

Reviewer #2: No
